# Plurisubharmonic Interpolation and Plurisubharmonic Geodesics

Alexander Rashkovskii 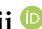

Department of Mathematics and Physics, University of Stavanger, 4036 Stavanger, Norway;
alexander.rashkovskii@uis.no

**Abstract:** We give a short survey on plurisubharmonic interpolation, with a focus on the possibility of connecting two given plurisubharmonic functions by plurisubharmonic geodesics.

**Keywords:** plurisubharmonic functions; pluricomplex Green function; energy functional; Monge–Ampère operator; plurisubharmonic geodesic; Cegrell class

## 1. Introduction

In a model example of the classical Calderón's complex interpolation theory [1] (see also [2,3]), two Banach spaces $X_j = (\mathbb{C}^n, \|\cdot\|_j)$ $(j = 0, 1)$ are interpolated by intermediate Banach spaces $X_\zeta = (\mathbb{C}^n, \|\cdot\|_\zeta)$ for $\zeta \in S = \{\zeta = \sigma + i\tau : 0 < \sigma < 1\} \subset \mathbb{C}$ with the norms

$$\|w\|_\zeta = \|w\|_\sigma = \inf\left\{\max_{j=0,1} N_j(f) : f \in \mathcal{F}, f(\zeta) = w\right\},$$

where $\mathcal{F}$ is the family of mappings $f : S \to \mathbb{C}^n$, bounded and analytic in $S$, continuous up to the boundary, $f(\sigma + i\tau) \to 0$ as $\tau \to \pm\infty$, and $N_j(f) = \sup_{\tau \in \mathbb{R}} \|f(j + i\tau)\|_j$, $j = 0, 1$.

Its plurisubharmonic version was considered in [4] as follows. Given two plurisubharmonic functions $u_0, u_1$ in a bounded domain $D$ of $\mathbb{C}^n$, find a plurisubharmonic function $u$ in $D \times \mathcal{A} \subset \mathbb{C}^{n+1}$ with $\mathcal{A} = \{\zeta : 0 < \log|\zeta| < 1\} \subset \mathbb{C}$, whose boundary values on $\mathcal{A}_j = \{\log|\zeta| = j\}$ coincide with $u_j$ $(j = 0, 1)$ (the annulus $\mathcal{A}$ was used instead of the strip $S$ in order to stick to the standard setting of the Dirichlet problem for the complex Monge–Ampère operator in bounded domains).

More precisely, denote

$$W(u_0, u_1) = \{v \in \mathrm{PSH}(D \times \mathcal{A}) : \limsup_{\zeta \to \mathcal{A}_j} v(\cdot, \zeta) \leq u_j(\cdot), \, j = 0, 1\}. \tag{1}$$

When both $u_0$ and $u_1$ are bounded above, this set is not empty (it contains $u_0 + u_1 - M$ for a constant $M$ big enough). Note also that for any $v \in W(u_0, u_1)$, the function $\sup\{v(z, \xi) : |\xi| = |\zeta|\}$ belongs to $W(u_0, u_1)$ as well.

Let $\hat{u} = \sup\{v \in W(u_0, u_1)\}$. It is a plurisubharmonic function in $D \times \mathcal{A}$, depending on $z$ and $|\zeta|$ and so, in particular, it is a convex function of $\log|\zeta|$.

**Definition 1.** *A family $v_t(z)$ on $\mathbb{D} \times (0, 1)$ is a subgeodesic for $u_0, u_1$ if $v_{\log|\zeta|}(z) \in W(u_0, u_1)$.*

*The largest subgeodesic, $u_t$, is the geodesic. In other words, $u_t(z) = u_{\log|\zeta|}(z) = \hat{u}(z, \zeta)$ for $t = \log|\zeta|$.*

This shows that the plurisubharmonic interpolation problem is equivalent to finding a (sub)geodesic passing through the two given plurisubharmonic functions, which we will call here the *geodesic connectivity problem*.

The origins of the plurisubharmonic geodesics lie in studying Kähler metrics on compact complex manifolds $(X, \omega)$. Starting with [5], a notion of geodesics in the spaces

of such metrics has been playing a prominent role in Kähler geometry and has found a lot of applications; see, for example [6–9], and the bibliography therein. Considerable progress was made then by relating the metrics to quasi-psh functions on compact complex manifolds; see [10–23], and many others.

We would especially like to refer to [12,22] where the connectivity problem for quasi-plurisubharmonic functions on compact Kähler manifolds was for the first time treated in terms of *rooftop envelopes*; the approach was developed then in [15–19,24,25] and other recent papers. A nice overview of this activity can be found in [26].

For the local setting of plurisubharmonic functions on bounded domains, the geodesics were considered in the (unpublished) preprint [27] and, independently, in [28,29], and then in [4,30–33]. Comparing with the compact manifold case, two main difficulties arise. First, a plurisubharmonic function can have its singularities on the boundary of the domain, and theory of boundary behavior of such plurisubharmonic functions is at the moment underdeveloped. Another issue is the lack of control over the total Monge–Ampère mass of plurisubharmonic functions and of monotonicity for its non-pluripolar part, the central tools used in the compact case.

In this paper, we survey the local theory results; also, we present a few new results on solvability of the connectivity problem in the Cegrell class $\mathcal{F}$.

We start with the simplest case of interpolation of bounded plurisubharmonic functions and functions from the Cegrell class $\mathcal{E}_0$ (Sections 1 and 2) and applications to set interpolation by means of relative extremal functions (Sections 3–5). In Section 6, we extend these results to the Cegrell class $\mathcal{F}_1$. The main tool for interpolation of unbounded functions is the rooftop technique, which we present in Section 7. A general setting of the connectivity problem is given in Section 8, and we introduce one of the main objects of the theory, residual plurisubharmonic functions, in Section 9. We formulate a fundamental Rooftop Equality conjecture and confirm it for functions from the class $\mathcal{F}$ in Section 10, and we apply it to solving the connectivity problem for certain cases in Section 11. Finally, in Section 12, we mention a few other directions in the field of plurisubharmonic interpolation and geodesics.

The set of all plurisubharmonic (*psh* for short) functions in a domain $\Omega$ will be denoted by $\mathrm{PSH}(\Omega)$, and its subset of negative functions is $\mathrm{PSH}^-(\Omega)$. For basics on psh functions, we refer to [34,35]. A nice (and comprehensive) presentation of the theory of Cegrell classes can be found in [36].

## 2. Bounded Psh Functions and Class $\mathcal{E}_0$

Let $D$ be a bounded domain in $\mathbb{C}^n$ and let $u_0, u_1 \in \mathrm{PSH}(D) \cap L^\infty(D)$. We consider the class $W(u_0, u_1)$ given by (1) and define its upper envelope $\widehat{u}$ and geodesic $u_t$ as described in Introduction. We define the subgeodesic $V_t := \max\{u_0 - Mt, u_1 - M(1-t)\}$, where $M = \|u_0 - u_1\|_\infty$, and obtain

$$V_t \leq u_t \leq (1-t)u_0 + tu_1, \quad 0 < t < 1 \tag{2}$$

(the second inequality being a consequence of convexity of $u_t$ in $t$), which implies that $u_t \to u_j$, uniformly on $D$, as $t \to j \in \{0,1\}$. This means that, in the bounded case, the geodesic always exists and attains the boundary values in a very nice sense.

When $u_j = 0$ on $\partial D$, one has control on the regularity of the upper envelope $\widehat{u}$ of $W(u_0, u_1)$ and, therefore, on $u_t$:

**Theorem 1** ([29]). *If $u_j \in C^{1,1}(D)$, then $u_t \in C^{1,1}(D)$ both in z and t.*

Subgeodesics and geodesics of psh functions have some special properties when the functions belong to a specific class—namely, to the Cegrell class $\mathcal{F}_1$. Here we start, however, with another Cegrell class, $\mathcal{E}_0$.

Assume $D$ is a bounded *hyperconvex* domain, i.e., there exists a negative psh function $\rho$ exhausting $D$: $D_c = \{z : \rho(z) < c\} \Subset D$ for all $c < 0$ and $\cup_{c<0}D_c = D$. The Cegrell class $\mathcal{E}_0(D)$ is formed by all bounded psh functions $u$ in $D$ that have zero boundary value on $\partial D$ ($u(z) \to 0$ as $z \to \partial D$) and finite total Monge–Ampère mass: $\int_D (dd^c u)^n < \infty$.

Note that if $u_0, u_1 \in \mathcal{E}_0(D)$, then $u_t \in \mathcal{E}_0(D)$ for any $t$ as well.

Consider the *energy functional*

$$\mathbf{E}(u) = \int_D u(dd^c u)^n.$$

By integration by parts,

$$\mathbf{E}(u) - \mathbf{E}(v) = \int_D (u-v) \sum_{k=0}^n (dd^c u)^k \wedge (dd^c v)^{n-k}, \quad u, v \in \mathcal{E}_0(D).$$

**Theorem 2** ([28]). *Let $u, v \in \mathcal{E}_0(D)$ satisfy $u \le v$. Then*

1.  $\mathbf{E}(u) \le \mathbf{E}(v)$;
2.  *if $\mathbf{E}(u) = \mathbf{E}(v)$, then $u = v$.*

The energy functional has the following remarkable properties.

**Theorem 3** ([27,28]). *Let $u_0, u_1 \in \mathcal{E}_0(D)$. Then:*

1.  *For any subgeodesic $v_t \in \mathcal{E}_0(D)$, $\mathbf{E}(v_t)$ is a convex function of $t$;*
2.  *For a subgeodesic $v_t \in \mathcal{E}_0(D)$, $t \mapsto \mathbf{E}(v_t)$ is linear if and only if $v_t$ is a geodesic for some $u_0, u_1 \in \mathcal{E}_0(D)$.*

From this, one can deduce the following uniqueness result.

**Theorem 4** ([28]). *If $u_0, u_1 \in \mathcal{E}_0(D)$ satisfy $\int_D u_0 (dd^c u_0)^k \wedge (dd^c u_1)^{n-k} = \mathbf{E}(u_1)$ for $k = 0, \ldots, n$, then $u_0 = u_1$.*

This works also for larger classes of psh functions; however, $\mathcal{E}_0$ suffices for an application to plurisubharmonic interpolation of certain measures and sets.

### 3. Relative Extremal Functions

Let $K \Subset D$. Recall that the *relative extremal function* of $K$ w.r.t. $D$ is

$$\omega_K = \omega_{K,D} = \sup{}^* \{u \in \mathrm{PSH}^-(D) : u|_K \le -1\}$$

(here $\sup^*$ is the upper semicontinuous regularization of sup). This is a function from $\mathcal{E}_0(D)$, *maximal* outside $K$: $(dd^c \omega_K)^n = 0$ on $D \setminus K$. The *relative capacity* of $K$ (w.r.t. $D$) can be defined as

$$\mathrm{Cap}\,(K) = (dd^c \omega_K)^n(K).$$

Let $u_t$ be the geodesic between $u_j = \omega_{K_j}$, $j = 0, 1$. Consider the sets

$$\widehat{K}_t = \{z : u_t(z) = -1\}, \quad 0 < t < 1, \tag{3}$$

then $-\mathbf{E}(u_t) \ge \mathrm{Cap}\,(\widehat{K}_t)$ while $\mathbf{E}(\omega_K) = -\mathrm{Cap}\,(K)$, so we obtain

**Theorem 5** ([28]). *For any $K_j \Subset D$,*

$$\mathrm{Cap}\,(\widehat{K}_t) \le (1-t)\,\mathrm{Cap}\,(K_0) + t\,\mathrm{Cap}\,(K_1).$$

One might expect that the geodesic interpolations $u_t$ of $\omega_{K_j}$ are $\omega_{\widehat{K}_t}$, which would replace the inequality in this theorem by equality. It turns out to be false, even in the simplest examples (see also Theorem 9):

**Example 1.** *Let* $n = 1$, $D = \mathbb{D}$, $K_0 = \{z : |z| \le e^{-1}\}$, $K_1 = \{z : |z| \le e^{-2}\}$. *Then* $\widehat{K}_t = \{z : |z| \le e^{-1-t}\}$, *while*

$$u_t(z) = \max\left\{\log|z|, \frac{\log|z| + t - 1}{2}, -1\right\}$$

*is not a relative extremal function. We have* $\operatorname{supp} dd^c u_j = \partial K_j = \{z : \log|z| = -1 - j\}$ *and* $\operatorname{supp} dd^c u_t = \{z : \log|z| = -1 \pm t\} = \partial \widehat{K}_t \cup \{z : \log|z| = -1 + t\}$.

While the sets $\widehat{K}_t$, as defined, could depend on the choice of the domain $D$, this is not actually true, at least in the case when $K_j$ are *polynomially convex*, which means that

$$K_j = \{z \in \mathbb{C}^n : |P(z)| \le \|P\|_{K_j} \ \forall P \in \mathcal{P}\},$$

$\mathcal{P}$ being the collection of all polynomials. Namely, they are sections of certain holomorphic hulls of the set $K^{\mathcal{A}} := (K_0 \times \mathcal{A}_0) \cup (K_1 \times \mathcal{A}_1) \subset \mathbb{C}^{n+1}$ with respect to functions holomorphic in $\mathbb{C}^n \times (\mathbb{C} \setminus 0)$:

**Theorem 6** ([4]). *If* $K_j$ *are compact and polynomially convex, then, for any* $\zeta \in \mathcal{A}$,

$$\widehat{K}_{\log|\zeta|} = \{z \in \mathbb{C}^n : |f(z,\zeta)| \le \|f\|_{K^{\mathcal{A}}} \quad \forall f \in \mathcal{O}(\mathbb{C}^n \times (\mathbb{C} \setminus 0))\}.$$

## 4. Toric Case

More can be said if $K_j$ are closures of complete, logarithmically convex, multicircled (Reinhardt) domains, which means that $y \in K_j$ provided $z \in K_j$ and $|y_i| \le |z_i|$ for all $i$, and

$$\operatorname{Log} K_j := \{s \in \mathbb{R}^n : \exp s = (e^{s_1}, \ldots, e^{s_n}) \in K_j\}$$

are convex subsets of $\mathbb{R}^n$. When, in addition, the domain $D$ is multicircled, the functions $\omega_{K_j}$ are *toric* (multicircled), and so are the geodesics $u_t$.

Any toric plurisubharmonic function $u(z)$ on $D$ can be identified with its *convex image*: the convex function $\breve{u}(s) = u(\exp s)$ on $\operatorname{Log} D \subset \mathbb{R}^n$, increasing in each variable $s_j$. In addition, $\breve{u}_t$ is a convex function on $\operatorname{Log} D \times (0,1) \subset \mathbb{R}^{n+1}$.

The geodesics between toric psh functions have an easy description:

**Theorem 7** ([31,37]). *The convex image* $\breve{u}_t$ *of any toric geodesic* $u_t$ *is given by*

$$\breve{u}_t = \mathcal{L}[(1-t)\mathcal{L}[\breve{u}_0] + t\mathcal{L}[\breve{u}_1]],$$

*where* $\mathcal{L}$ *is the Legendre transform,*

$$\mathcal{L}[f](y) = \sup_{x \in \mathbb{R}^n_-} \{\langle x, y \rangle - f(x)\}.$$

We can assume $D = \mathbb{D}^n$, the unit polydisk. By [38],

$$\omega_{K_j}(z) = \sup_{a \in \mathbb{R}^n_+} \frac{\sum a_k \log|z_k|}{|h_{L_j}(a)|}, \quad z \in \mathbb{D}^n \setminus K_j,$$

where $L_j = \operatorname{Log} K_j \subset \mathbb{R}^n_- = \{s \in \mathbb{R}^n : s_j \leq 0, \ 1 \leq j \leq n\}$ and

$$h_{L_j}(a) = \sup_{s \in L_j} \langle a, s \rangle, \quad a \in \mathbb{R}^n,$$

is the support function of $L_j$. Then $\mathcal{L}[\check{\omega}_{K_j}] = \max\{h_{L_j} + 1, 0\}$, which gives an explicit formula for the geodesic $u_t$ of $u_j = \omega_{K_j}$:

$$\check{u}_t = \mathcal{L}[(1-t)\max\{h_{L_0} + 1, 0\} + t\max\{h_{L_1} + 1, 0\}].$$

As a consequence, we obtain

**Theorem 8** ([31]). *Let $u_t$ be the geodesic between $u_j = \omega_{K_j}$ for complete, logarithmically convex Reinhardt sets $K_j \Subset \mathbb{D}^n$. Then the sets $\widehat{K}_t$ defined by (3) are the geometric means of $K_j$: $\widehat{K}_t = K_0^{1-t} K_1^t$; in other words, $\operatorname{Log} \widehat{K}_t = (1-t)\operatorname{Log} K_0 + t\operatorname{Log} K_1$.*

One can show that, in the toric case, there can never be an equality in Theorem 5, unless the geodesic is constant:

**Theorem 9** ([31]). *In the conditions of Theorem 8, if there exists $0 < t_0 < 1$ such that $u_{t_0} = \omega_{K_{t_0}}$, then $u_0 = u_1$.*

Recall that volumes of convex combinations $K(t) = (1-t)K_0 + tK_1$ of convex bodies $K_j \subset \mathbb{R}^n$ satisfy the Brunn–Minkowski inequality

$$\operatorname{Vol}(K(t)) \geq \operatorname{Vol}(K_0)^{1-t}\operatorname{Vol}(K_1)^t;$$

in other words, volumes of $K(t)$ are *logarithmically concave*. The same is true for the multiplicative combinations $\widehat{K}_t = K_0^{1-t} K_1^t$ of convex Reinhardt bodies $K_j \subset \mathbb{C}^n$ [39].

The geodesic interpolation gives us a *reverse estimate*: the (usual) *convexity* of the capacities

$$\operatorname{Cap}(K_t) \leq (1-t)\operatorname{Cap}(K_0) + t\operatorname{Cap}(K_1)$$

for logarithmically convex Reinhardt bodies $K_j$.

## 5. Weighted Extremal Functions

One can obtain a stronger relation between the capacities if the functions $u_j = \omega_{K_j}$ are replaced with weighted ones $u_j = c_j\omega_{K_j}$ for some $c_j > 0$.

Let $u_t^c$ be the corresponding geodesic. The sets $\widehat{K}_t$ are now to be defined as

$$\widehat{K}_t^c = \{z \in \Omega : u_t^c(z) = m_t\}, \quad 0 < t < 1,$$

where $m_t = \min\{u_t^c(z) : z \in \Omega\}$.

It can be shown that $m_t$ is a concave function and

$$|m_t| \leq c_t := (1-t)c_0 + tc_1.$$

Moreover, if $K_0 \cap K_1 \neq \varnothing$, then $|m_t| = c_t$.

**Theorem 10** ([32]). *Let $K_j \Subset D$ be polynomially convex, $K_0 \cap K_1 \neq \varnothing$, and let the weights $c_j$ be chosen such that*

$$c_0^{n+1}\operatorname{Cap}(K_0) = c_1^{n+1}\operatorname{Cap}(K_1).$$

*Then*

$$\left(\operatorname{Cap}(\widehat{K}_t^c)\right)^{-\frac{1}{n+1}} \geq (1-t)\left(\operatorname{Cap}(\widehat{K}_t^c)\right)^{-\frac{1}{n+1}} + t\left(\operatorname{Cap}(\widehat{K}_t^c)\right)^{-\frac{1}{n+1}}.$$

A little drawback of this result is that the sets $\widehat{K}_t^c$ depend on the parameters $c_j$. It turns out not to be the case in the toric situation, and the capacity inequality becomes the one on concavity of the function $\left(\mathrm{Cap}\,(\widehat{K}_t)\right)^{-\frac{1}{n+1}}$:

**Theorem 11** ([32])**.** *If $K_j \subset \mathbb{D}^n$ are closures of complete, logarithmically convex Reinhardt domains, then $\widehat{K}_t^c = \widehat{K}_t = K_0^{1-t} K_1^t$ for any weights $c_j > 0$. Furthermore, the geodesic $u_\tau^c$ equals $c_\tau\,\omega_{K_\tau}$ for some $\tau \in (0,1)$ if and only if $K_1^{c_0} = K_0^{c_1}$ (that is, $c_0 \operatorname{Log} K_1 = c_1 \operatorname{Log} K_0$) and so, $u_t^c = c_t\,\omega_{K_t}$ for all $t$.*

Since the concavity of $v^{-1}(t)$ implies convexity of $v(t)$ and since the function $x \mapsto x^{1+\frac{1}{n}}$ is convex, the conclusion of Theorem 11 is stronger than convexity of $(\mathrm{Cap}\,(\widehat{K}_t))^{\frac{1}{n}}$, and the latter is equivalent to logarithmic convexity of $\mathrm{Cap}\,(\widehat{K}_t)$. This implies

**Corollary 1** ([32])**.** *In the conditions of Theorem 11,*

$$\mathrm{Cap}\,(\widehat{K}_t) \le \mathrm{Cap}\,(K_0)^{1-t}\mathrm{Cap}\,(K_1)^t.$$

## 6. Geodesics in $\mathcal{F}_1$

To obtain the above results on geodesics, including the linearity of the energy functional **E**, extended to larger classes of psh functions, one should stick to those where the functional is still finite. This leads to considering *Cegrell's energy classes*, of which the simplest one is the class $\mathcal{F}_1$.

Let $D$ be a bounded hyperconvex domain in $\mathbb{C}^n$.

**Definition 2** ([40,41])**.** *The class $\mathcal{F}(D)$ is formed by all $u \in \mathrm{PSH}^-(D)$ that are limits of decreasing sequences $u_N \in \mathcal{E}_0(D)$ such that*

$$\sup_N \int_D (dd^c u_N)^n < \infty.$$

*If, in addition, $\sup_N \int_D |u_N|(dd^c u_N)^n < \infty$, then $u \in \mathcal{F}_1(D)$. A function $v \in \mathrm{PSH}^-(D)$ belongs to $\mathcal{E}(D)$ if for any $D' \Subset D$ there exists $u \in \mathcal{F}(D)$ coinciding with $v$ on $D'$.*

For any $u \in \mathcal{F}_1(D)$, $(dd^c u)^n = \lim_{N \to \infty}(dd^c u_N)^n$, $u(dd^c u)^n = \lim_{N \to \infty} u_N(dd^c u_N)^n$, and $\mathbf{E}(u) = \lim_{N \to \infty} \mathbf{E}(u_N)$. Then, given $u_j \in \mathcal{F}_1(D)$, we approximate them by $u_{j,N} \in \mathcal{E}_0(D)$, find the geodesics $u_{t,N}$, and then we can look at $u_t = \lim u_{N,t}$ as $N \to \infty$.

A piece missing from the bounded case is an argument for $u_t$ to converge to $u_j$ as $t \to j$, since one cannot have $L^\infty$-bounds now. Instead, a rooftop technique can be used. Since it will be playing a central role in the rest of the exposition, we will present it in the next section, while here we just state a result on $\mathcal{F}_1$-geodesics based on that technique.

Let $P(u,v)$ be the *rooftop envelope* (the largest psh minorants of $\min\{u,v\}$). Then, for any $u_0, u_1 \in \mathrm{PSH}^-(D)$ and any $C \ge 0$, the curve $w_{C,t} = P(u_0, u_1 + C) - Ct$, $0 < t < 1$, is evidently a subgeodesic, and it plays the role of the subgeodesics $V_t$ (2) from the bounded case.

**Theorem 12** ([28])**.** *Let $u_0, u_1 \in \mathcal{F}_1(D)$. Then*

1. *for any subgeodesic $v_t \subset \mathcal{F}_1(D)$, the function $t \mapsto \mathbf{E}(v_t)$ is convex;*
2. *for the geodesic $u_t$, the function $t \mapsto \mathbf{E}(u_t)$ is affine;*
3. *$u_t \to u_j$ in capacity as $t \to j \in \{0,1\}$: $\forall \epsilon > 0$, $\mathrm{Cap}\,\{|u_t - u_j| \ge \epsilon\} \to 0$.*

Uniqueness Theorems 2 and 4 also remain true for $u, v \in \mathcal{F}_1(D)$ [28].

## 7. Rooftop Envelopes

Rooftop envelopes were explicitly introduced in [22] for quasi-psh functions on compact Kähler manifolds, and then the technique was developed in [12,14,15,21,23,42] and others. In the local context, they were considered first in [28] for functions in the Cegrell class $\mathcal{F}_1$ and then in [33] for arbitrary psh functions, bounded from above.

The *rooftop envelope* $P(u,v)$ of bounded above functions $u$ and $v$ is the largest psh minorant of $\min\{u,v\}$. Since $P(u,v) \geq u + v - M$ for some $M \geq 0$, $P(u,v) \not\equiv -\infty$.

As follows from Prop. 3.3 in [12] (see also Lemma 3.7 in [16]),

$$\text{NP}(dd^c[P(u,v)])^n \leq \mathbb{1}_{\{P(u,v)=u\}}\text{NP}(dd^c u)^n + \mathbb{1}_{\{P(u,v)=v\}}\text{NP}(dd^c v)^n, \tag{4}$$

where $\text{NP}(dd^c w)^n$ is the *non-pluripolar Monge–Ampère operator* in the sense of [43]: for Borel sets $E$,

$$\text{NP}(dd^c w)^n = \lim_{j\to\infty} \mathbb{1}_{E\cap\{w>-j\}}(dd^c \max\{w,-j\})^n.$$

In particular, $P(u,v)$ satisfies $(dd^c[P(u,v)])^n = 0$ on $\{-\infty < P(u,v) < \min\{u,v\}\}$.

While $P(u,v_j)$ decreases to $P(u,v)$ when $v_j$ decreases to $v$, its behavior for increasing $v_j$ can be more complicated, provided $v_j$ are unbounded from below.

**Example 2.** *Let $D = \mathbb{D}^n$, $u = 0$, $v_j = \max_k \log|z_k| + j$. Then $\min\{u,v_j\}$ increase, as $j \to \infty$, to the function $\hat{h}$ equal to $0$ outside the origin and $\hat{h}(0) = -\infty$, while $P(u,v_j) = v_0$ for all j.*

This observation is a particular case of how the rooftop envelopes $P(u,v+C)$ behave when $C \to \infty$. Denote

$$\sup_C{}^* P(u,v+C) = P[v](u),$$

the *asymptotic rooftop*, or *asymptotic envelope* of $u$ with respect to the singularity of $v$.

**Lemma 1** ([28]). *If $u,v \in \mathcal{F}_1(D)$, then $P[v](u) = u$.*

**Remark 1.** *This actually implies the connectivity of any $u_0, u_1 \in \mathcal{F}_1(D)$; see Theorem 13.*

One can ask if this works for any negative psh functions. The rest of the paper will be actually devoted to this question.

## 8. Geodesics on $\text{PSH}^-(D)$

Any $u \in \text{PSH}^-(D)$ is the limit of a decreasing sequence $u_N \in \mathcal{E}_0(D)$ [44]. So, for any pair $u_j \in \text{PSH}^-(D)$, $j = 0,1$, we repeat what we did for $\mathcal{F}_1$: approximate $u_j$ by $u_{j,N}$, connect them by the geodesics $u_{t,N}$, and then obtain the 'geodesic' $u_t$ as the limit of $u_{t,N}$ as $N \to \infty$.

The crucial question is if $u_t$ connects $u_j$. More precisely: *Does $u_t$ converge to $u_j$, in any sense, as $t \to j \in \{0,1\}$?*

**Example 3.** *Let $D = \mathbb{D}$, $u_0 = 0$, $u_1 = \log|z| \in \mathcal{F}(\mathbb{D}) \setminus \mathcal{F}_1(\mathbb{D})$.*

*For any $N > 0$, the function $u_{N,t} = \max\{u_1, -Nt\}$ is the geodesic between $u_0$ and $u_{1,N} = \max\{u_1, -N\}$. Therefore, $u_t \equiv u_1 = \log|z|$ is not passing through $u_0$.*

*The same works for $u_0 = 0$ and $u_1 = G_a \in \mathcal{F}(\mathbb{D}) \setminus \mathcal{F}_1(\mathbb{D})$, the pluricomplex Green function with pole at $a \in D \subset \mathbb{C}^n$, $n > 1$.*

Even more striking is

**Example 4.** *Let $u_j = G_{a_j}$ be the (pluricomplex) Green functions with different poles. Then $u_t$ does not exceed the geodesic between $G_{a_0}$ and $0$, that is, by Example 3, $G_{a_0}$, and, by the same argument, it does not exceed $G_{a_1}$. Therefore, it does not exceed (actually, equals) $P(G_{a_0}, G_{a_0}) = G_{a_0,a_1}$, the Green functions with two logarithmic poles.*

*In this case, we obtain a 'geodesic' that does not pass through any of the endpoints.*

So, there are functions that cannot be connected by geodesics, the obstacle being that they have different 'strong' singularities. This sets the following

*Geodesic connectivity problem*: *What pairs $u_0, u_1 \in \mathrm{PSH}^-(D)$ can be connected by a psh (sub)geodesic?*

The problem in the compact setting (quasi-psh functions on compact Kähler manifolds) was handled in [12] in terms of *asymptotic envelopes*, and this easily adapts to the local case:

**Theorem 13** ([33]). *Let $u_0, u_1 \in \mathrm{PSH}^-(D)$, then the geodesic $u_t$ converges to $u_0$ in $L^1_{loc}(D)$ (and in capacity) as $t \to 0$ if and only if $P[u_1](u_0) = u_0$.*

As we have already seen, the possible obstacles can arise only from the singularities of $u_0$ and $u_1$. When the singularities are equivalent, $u_0 \asymp u_1$, which means

$$u_0(z) - A \le u_1(z) \le u_0(z) + A$$

for some $A > 0$ and all $z \in D$, we have both $P[u_1](u_0) = u_0$ and $P[u_0](u_1) = u_1$ and so, the geodesic connects the data functions.

This, however, does not cover Theorem 12 because functions from $\mathcal{F}_1$ need not have equivalent singularities. Then one should look for a coarser equivalency of singularities.

In [30], it was proved that two toric psh functions in $\mathbb{D}^n$ with isolated singularities at 0 can be connected if and only if all their *directional Lelong numbers* coincide. This means that the main terms of their singularities are the same. The proof was based on relating toric psh functions to convex ones (as in Section 4) and then using the technique of convex analysis. This will not work for arbitrary psh functions, so one should find another way to single out such 'main terms' of the singularities.

## 9. Residual Function

Most of the contents of this section are taken from [33].

**Definition 3.** *Given $\phi \in \mathrm{PSH}^-(D)$, its residual function is*

$$g_\phi = g_{\phi,D} = P[\phi](0) = \sup_{C \ge 0}{}^* P(\phi + C, 0)\} \in \mathrm{PSH}^-(D).$$

*Equivalently, $g_\phi$ is the (u.s.c. regularization) of the upper envelope of the class of all functions $u \in \mathrm{PSH}^-(D)$ with singularities at least as strong as that of $\phi$, meaning that $u \le \phi + C$ for some $C \in \mathbb{R}$.*

The function $g_\phi$ is determined by the asymptotic behavior of $\phi$ near its singularities, and it is a candidate for the main term of the asymptotic of the singularity of $\phi$.

**Example 5.** *If $\phi(z) \asymp \log|z - a|$, $a \in D$, then $g_\phi = G_a$, the pluricomplex Green function with pole at $a$.*

**Example 6.** *More generally, if $\phi(z) \asymp \sum c_k \log|z - a_k|$, then $g_\phi$ is the weighted multipole pluricomplex Green function.*

**Example 7.** *If $\phi$ is toric and $D = \mathbb{D}^n$, then $g_\phi$ coincides with its indicator $\Psi_u$ [45] defined in [46,47] as the toric psh function in $\mathbb{D}^n$ whose convex image $\psi_u(s) = \Psi_u(\exp s)$ in $\mathbb{R}^n_-$ is given by the directional Lelong numbers $\nu_{u,a}$ of $u$ at 0 in the directions $a \in \mathbb{R}^n_+$:*

$$\psi_u(s) = -\nu_{u,-s}, \quad s \in \mathbb{R}^n_-.$$

The next two examples deal with functions with non-isolated singularities.

**Example 8.** *Let $\phi(z) \asymp \log|z_1|$ and $D = \mathbb{D}^n$, then $g_\phi(z) = \log|z_1|$.*

**Example 9.** *Let $\phi(z) \asymp \log|z_1|$ and $D = \mathbb{B}^n$, then [48]*

$$g_\phi(z) = \log \frac{|z_1|}{\sqrt{1 - |z'|^2}}.$$

These two are particular cases of *Green functions with poles along complex spaces*; given an ideal $\mathcal{I} = \langle f_1, \ldots, f_N \rangle \subset \mathcal{O}(D)$ with bounded generators, $G_\mathcal{I}$ is the upper envelope of $u \in \mathrm{PSH}^-(D)$ such that $u \leq \log|f| + O(1)$ [49]. Note that in this definition, the asymptotics of $u$ and $\log|f|$ are related only locally, not uniformly in $D$, so their equality to the corresponding residual functions is not a trivial fact.

A bit different are the next two examples dealing with boundary singularities.

**Example 10.** *If $D = \mathbb{D} \subset \mathbb{C}$ and $\phi = -\mathcal{P}_a$, the negative Poisson kernel with pole at $a \in \partial\mathbb{D}$, then $g_\phi = -\mathcal{P}_a$.*

**Example 11.** *More generally, if $D \subset \mathbb{C}^n$ is strongly pseudoconvex with smooth boundary, then for any $\zeta \in \partial D$ there exists the pluricomplex Poisson kernel $\Omega_\zeta \in \mathrm{PSH}^-(D)$ which satisfies $(dd^c\Omega_\zeta)^n = 0$ in $D$, is continuous in $\overline{D} \setminus \{\zeta\}$, equal to $0$ on $\partial D \setminus \{\zeta\}$, and such that $\Omega_\zeta(z) \approx -|z - \zeta|^{-1}$ as $z \to \zeta$ nontangentially [50,51].*

*We have $P(\Omega_\zeta + C, 0) = \Omega_\zeta$ for any $C > 0$ and so, $g_{\Omega_\zeta} = \Omega_\zeta$. When $D = \mathbb{B}^n$,*

$$\Omega_\zeta(z) = \frac{|z|^2 - 1}{|1 - \langle z, \zeta \rangle|^2}.$$

In the general case, the picture can be much more complicated. Since the singularities can lie both inside the domain and on its boundary, we call $g_\phi$ *the Green–Poisson residual function* of $\phi$ for the domain $D$.

By properties of rooftops, $(dd^c P(\phi + C, 0))^n = 0$ on $\{\phi > -C\}$, so the non-pluripolar Monge–Ampère current of $g_\phi$ is zero.

The boundary values of $g_\phi$ need not be zero (outside the unbounded locus of $\phi$); see Example 8. However, they are zero there if the domain is B-regular (i.e., each boundary point possesses a strong psh barrier [52]).

By the *unbounded locus of $u \in \mathrm{PSH}^-(D)$* we mean the set $L_u$ of all points $z \in \overline{D}$ such that $u$ is not bounded in $D \cap U_z$ for any neighborhood $U_z$ of $z$.

A very important property we believe the residual functions have is their *idempotency*:

$$g_{g_u} = g_u.$$

At the moment, however, it is known to hold only under some assumptions on $u$. To present them, we need the following

**Definition 4.** *$u \in \mathrm{PSH}^-(D)$ has small unbounded locus if there exists $v \in \mathrm{PSH}^-(D)$, $v \not\equiv -\infty$, such that $v^* = -\infty$ on $L_u$; here, for any $\zeta \in \overline{D}$, $v^*(\zeta) = \limsup_{z \to \zeta} v(z)$.*

This differs from the definition of small unbounded locus used in [33], where the requirement was *pluripolarity* of $L_u$, that is, existence of a function $V \not\equiv -\infty$ which is psh in a neighborhood of $\overline{D}$ and $V(\zeta) = -\infty$ for all $\zeta \in L_u$. The present definition does not change $L_u \cap D$, while it allows the boundary part $L_u \cap \partial D$ to be much bigger than pluripolar; such sets are called *b-pluripolar* [53]. For example, a compact set $K \subset \partial\mathbb{D} \subset \mathbb{C}$ is b-pluripolar if and only if it is of zero Lebesgue measure.

The following was proved in [33], and it can be checked that the proof of assertions (i) and (ii) with the new definition of small unbounded locus remains unchanged.

**Theorem 14.** *Let $u \in \mathrm{PSH}^-(D)$. Then $g_{g_u} = g_u$, provided one of the conditions is fulfilled:*

*(i)　u has small unbounded locus;*
*(ii)　the boundary function $\tilde{u}$ of $u \in \mathcal{E}(D)$, in the sense of Cegrell [54], has small unbounded locus;*
*(iii)　$u \in \mathcal{F}(D)$;*
*(iv)　$n = 1$ (i.e., $D \subset \mathbb{C}$).*

**Remark 2.** *In the one-dimensional case, the structure of residual functions is quite simple; if $u = \mathcal{G}_\mu + \mathcal{P}_\nu$ is the Green–Poisson representation of $u \in \mathrm{SH}^-(\mathbb{D})$, then $g_u = \mathcal{G}_{\mu_s} + \mathcal{P}_{\nu_s}$ with $\mu_s$, the restriction of the Riesz measure $\mu$ of $u$ to $\{u = -\infty\}$, and $\nu_s$, the singular part (with respect to the Lebesgue measure) of the boundary measure $\nu$ of $u$ [55,56].*

The idempotency of the residual functions has a lot of useful applications; for those concerning the geodesics, see Section 11.

The residual function $g_\phi$ keeps the main characteristics of singularities of $\phi$. Recall that the *Lelong number* $\nu_u(a)$ of a psh function $u$ at a point $a$ is the largest nonnegative number $\nu$ such that $u(z) \leq \nu \log|z - a| + O(1)$ near $a$, the *log canonical threshold* $c_u(a)$ of $u$ is the supremum of $c \geq 0$ such that $e^{-cu} \in L^2_{loc}(a)$, and the *multiplier ideal* $\mathcal{I}_u(a)$ is formed by all $f \in \mathcal{O}_a$ such that $|f|e^{-u} \in L^2_{loc}(a)$. Since they are all continuous for increasing sequences of psh functions, we obtain

**Theorem 15.** *For any $\phi \in \mathrm{PSH}^-(D)$ and $a \in D$, $\nu_{g_\phi}(a) = \nu_\phi(a)$, $c_{g_\phi}(a) = c_\phi(a)$, and $\mathcal{I}_{t g_\phi}(a) = \mathcal{I}_{t\phi}(a) \; \forall t > 0$.*

When $\phi \in \mathcal{F}(D)$, the residual function can actually be described explicitly.

**Theorem 16.** *If $(dd^c \phi)^n$ is well-defined, then so is $(dd^c g_\phi)^n$ and, furthermore,*

$$(dd^c g_\phi)^n = \mathbb{1}_{\{\phi = -\infty\}}(dd^c \phi)^n.$$

*In particular, if $\phi \in \mathcal{F}(D)$, then $g_\phi \in \mathcal{F}(D)$ and it is a unique solution to the equation $(dd^c u)^n = \mathbb{1}_{\{\phi = -\infty\}}(dd^c \phi)^n$ in the class $\mathcal{F}(D)$.*

**Remark 3.** *The uniqueness part here follows from [57].*

Functions from $\mathcal{F}(D)$ can have very large unbounded locus, and it can even coincide with $\overline{D}$. Nevertheless, the boundary value of any $u \in \mathcal{F}(D)$, in the sense of Cegrell, is zero, which means that the least psh majorant $H$ of $u$ in $D$ satisfying $(dd^c H)^n = 0$ is $H \equiv 0$.

More results on boundary behavior for functions from other Cegrell classes and their residual functions can be found in [33] (see also the last section of this paper).

## 10. Residual Functions and Asymptotic Rooftops

In the compact setting, the corresponding analog of the residual function is an ultimate tool for checking the connectivity of quasi-psh functions [16]. This was proved by using machinery of non-pluripolar Monge–Ampère operator, including the monotonicity property [58,59]. Unfortunately, that technique does not work in the local setting. In addition, functions from $\mathrm{PSH}^-(D)$ can have their singularities on the boundary, and theory of boundary behavior of such psh functions is still underdeveloped. That is why the results in the local theory are at the moment not that complete.

Let $\phi, \psi \in \mathrm{PSH}^-(D)$. Then $P[\phi](\psi) \leq g_\phi$, so

$$P[\phi](\psi) \leq P(g_\phi, \psi).$$

If $\phi \asymp g_\phi$ (i.e., $\phi \leq g_\phi \leq \phi + C$), then the *Rooftop Equality* holds:

$$P[\phi](\psi) = P(g_\phi, \psi) \quad \forall \psi \in \mathrm{PSH}^-(D). \tag{5}$$

Furthermore, it also holds for all $\phi, \psi \in \mathcal{F}_1(D)$ [28]. The following guess is then natural:

*Rooftop Equality conjecture:* (5) *holds for all* $\phi, \psi \in \mathrm{PSH}^-(D)$.

**Theorem 17** ([33]). *Let* $\phi \geq g_\phi + w$ *with some* $w \in \mathrm{PSH}^-(D)$ *such that* $g_w = 0$, *then the Rooftop Equality* $P[\phi](\psi) = P(g_\phi, \psi)$ *holds with any* $\psi \in \mathrm{PSH}^-(D)$.

Applying this to $w = \phi$, we obtain

**Corollary 2.** *The Rooftop Equality holds with any* $\psi \in \mathrm{PSH}^-(D)$ *if* $\phi$ *does not have strong singularities, i.e., if* $g_\phi = 0$.

In particular, this recovers the aforementioned result for $\mathcal{F}_1$ because $g_\phi = 0$ for all $\phi \in \mathcal{F}_1(D)$.

For the same reason, Corollary 2 proves the Rooftop Equality for the Cegrell class $\mathcal{F}^a(D)$ of functions $\phi \in \mathcal{F}(D)$ such that $(dd^c\phi)^n$ do not charge pluripolar sets.

It turns out, however, that Theorem 17 works also for the whole class $\mathcal{F}$:

**Corollary 3.** *If* $\phi \in \mathcal{F}(D)$, *then* $P[\phi](\psi) = P(g_\phi, \psi)$ *for any* $\psi \in \mathrm{PSH}^-(D)$.

**Proof.** By Cor. 4.15, 4.16 in [57], there exists a unique pair of functions $\phi_1, \phi_2 \in \mathcal{F}(D)$ such that $(dd^c\phi_1)^n = \mathbb{1}_{\{\phi > -\infty\}}(dd^c\phi)^n$, $(dd^c\phi_2)^n = \mathbb{1}_{\{\phi = -\infty\}}(dd^c\phi)^n$, and $\phi_1 + \phi_2 \leq \phi \leq P(\phi_1, \phi_2)$.

By Theorem 16, $(dd^c g_\phi)^n = \mathbb{1}_{\{\phi = -\infty\}}(dd^c\phi)^n$, so the uniqueness gives us $\phi_2 = g_\phi$. Since $\phi_1 \in \mathcal{F}^a(D)$, Theorem 17 with $w = \phi_1$ completes the proof. $\square$

**Corollary 4.** *If* $\phi, \psi \in \mathcal{F}(D)$, *then:*

(i)   $g_{\phi+\psi} = g_{g_\phi+g_\psi}$, *while the relation* $g_{\phi+\psi} = g_\phi$ *holds if and only if* $g_\psi = 0$;

(ii)  $g_{\max\{\phi,\psi\}} = g_{\max\{g_\phi, g_\psi\}}$, *while the relation* $g_{\max\{\phi,\psi\}} = g_\phi$ *holds if and only if* $g_\psi \leq g_\phi$;

(iii) $g_{P(\phi,\psi)} = P(g_\phi, g_\psi)$.

**Proof.** Assertions (i) and (ii), as well as the relations $P(g_\phi, g_\psi) = g_{P(g_\phi, g_\psi)}$ and $g_{P(\phi,\psi)} = g_{P[\phi](\psi)} = g_{P[\psi](\phi)}$, are proved in Prop. 3.7–3.9, Cor. 7.6 in [33] for $\phi, \psi$ with small unbounded loci; however, the only property used in the proofs was the idempotency of all the residual functions involved, which we have in our case of $\mathcal{F}(D)$. Then, by Corollary 3,

$$g_{P(\phi,\psi)} = g_{P(g_\phi, \psi)} = g_{P(g_\phi, g_\psi)} = P(g_\phi, g_\psi),$$

which proves (iii). $\square$

## 11. Geodesic Connectivity

Let $u_0, u_1 \in \mathrm{PSH}^-(D)$. Since $P[u_1](u_0) \leq P(g_{u_1}, u_0)$, the equality $P[u_1](u_0) = u_0$, which is equivalent to the condition $u_t \to u_0$ as $t \to 0$, implies $u_0 \leq g_{u_1}$, while the reverse implication would mean precisely the Rooftop Equality (5) for $\phi = u_1$ and $\psi = u_0$.

So, we have

**Theorem 18** ([33]). *Let* $u_0, u_1 \in \mathrm{PSH}^-(D)$ *and let the Rooftop Equality* (5) *be satisfied for* $\phi = u_j$, $j = 0, 1$. *Then* $u_t \to u_j$ *in* $L^1_{loc}(D)$ *(and in capacity) if and only if* $u_0 \leq g_{u_1}$ *and* $u_1 \leq g_{u_0}$. *If* $g_{u_j}$ *are idempotent, the two inequalities are equivalent to* $g_{u_0} = g_{u_1}$.

**Corollary 5.**

*(i)    Any two negative psh functions without strong singularities can be geodesically connected.*

*(ii)   Two functions from $\mathcal{F}(D)$ can be geodesically connected if and only if their residual functions coincide.*

*(iii)  No pair of psh functions with idempotent but different residual functions can be connected by a (sub)geodesic.*

*(iv)   Any negative psh function can be geodesically connected with its residual function.*

*(v)    Two negative subharmonic functions in $D \subset \mathbb{C}$ can be geodesically connected if and only if their residual functions coincide.*

**Remark 4.** *In the toric case, assertion (ii) covers the main result of [30].*

## 12. Other Results

We just mention some other directions related to psh geodesics on domains of $\mathbb{C}^n$.

1. One can consider *separately residual functions* $g_u^o$ and $g_u^b$ determined by the singularities of $u$ inside the domain and on its boundary, respectively [33]. For example, the Green function $G_{\mathcal{I}}$ with poles along a complex space given by an ideal sheaf $\mathcal{I} = \langle f_1, \ldots, f_k \rangle$, mentioned in Section 8, actually equals $g_{\log|f|}^o$; under some additional conditions on $\mathcal{I}$, it coincides with $g_{\log|f|}$, and in some situations, with $g_{\log|f|}^b$.

2. Residual functions $g_u$ for functions $u \in \mathrm{PSH}^-(D)$ with well-defined $(dd^c u)^n$ and possessing boundary values $\tilde{u}$ in the sense of Cegrell were considered in [33]. It was shown there that if $\int_D (dd^c u)^n < \infty$ and $\tilde{u}$ has small unbounded locus, then $g_u$ is idempotent and equal to $P(g_u^o, g_u^b)$.

3. Cegrell's energy classes can be considered for $m$-subharmonic functions on domains of $\mathbb{C}^n$, $1 \le m < n$ [60,61]. Geodesics for such functions, including the linearity of the corresponding energy functional, were studied in [62].

4. The Dirichlet problem for unbounded psh functions and its relation to the asymptotic rooftop construction were recently considered in [63–65].

5. The regularity of toric and convex geodesics was studied in [66].

**Funding:** This research received no external funding.

**Conflicts of Interest:** The authors declare no conflict of interest.

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
