# Peer review of "Plurisubharmonic Interpolation and Plurisubharmonic Geodesics"

_axioms, doi:10.3390/axioms12070671_

Round 1
Reviewer 1 Report
The article practically does not contain new results. However, the extensive
bibliography given in the bibliography may be of benefit to specialists dealing with the plurisubharmonic interpolation problem and the geodesic connectivity problem.
Therefore, I think that the article "Plurisubharmonic interpolation and
plurisubharmonic geodesics"of Alexander Rashkovskii deserves publication in the journal "Axioms".
Author Response
Thank you very much for the kind review.
Reviewer 2 Report
The paper is a survey on plurisubharmonic interpolation, quite exhaustive on the subject and well written.
There are some minor points I'd like to note:
1) Sometimes Cegrell is uncorrectly spelled Cegrel
2) l. 94: smth* should be sup* (same for smth, should be sup)
3) Thm 7: what is the "convex image"? It appears for the first time in the statement of the theorem. Seing the notation used in the lines preceding the theorem, it looks clear what it is, but it should be stated clearly.
Other than that minor points, the survey is weel wirtten and may be of interest to people working in the area.
Author Response
1) Fixed.
2) Fixed.
3) A formal definition of the term "convex image" is given now in the lines preceding Theorem 7.